# Two Novel Hydrate Salts of Norfloxacin with Phenolic Acids and Their Physicochemical Properties

**DOI:** 10.3390/antibiotics13090888

**Published:** 2024-09-14

**Authors:** Di Liang, Fei Li, Jingzhong Duan, Wei Sun, Xiaoyan Yu

**Affiliations:** School of Pharmaceutical Sciences, Jilin University, Changchun 130021, China; liangdi@jlu.edu.cn (D.L.); lif23@mails.jlu.edu.cn (F.L.); duanjz22@mails.jlu.edu.cn (J.D.); wsun@jlu.edu.cn (W.S.)

**Keywords:** antibacterial agents, norfloxacin, salt, dissolution studies, Hirshfeld surface

## Abstract

Norfloxacin (NORF) is a broad-spectrum quinolone that is widely utilized for the treatment of various bacterial infections and is considered one of the most commonly used fluoroquinolone antibiotics. However, NORF’s clinical utility is limited by its poor water solubility and relatively low oral bioavailability. This study presents an optimization and synergistic enhancement approach through salt/co-crystal, aiming to maximize the biopharmaceutical properties of NORF with the use of phenolic acid. Following this strategy, two new hydrate salts of NORF with phenolic acid, namely, NORF—3,5-DBA hydrate (salt **1**) and NORF—VA hydrate (salt **2**), were prepared and systematically confirmed. Two hydrate salts were produced by means of the slow evaporation crystallization method, and the structures were determined through single-crystal X-ray diffraction (SCXRD). Additionally, powder X-ray diffraction (PXRD), Fourier-transform infrared (FT-IR) spectroscopy, differential scanning calorimetry (DSC), thermogravimetric analysis (TGA), and high-performance liquid chromatography (HPLC) were applied to analyze the features of the two salts. The experimental results indicated that the formation of the two salts could enhance the solubility and improve the release behavior of NORF. Interestingly, the physicochemical properties of NORF were significantly improved as a result, leading to an enhancement in its antibacterial activity. This was demonstrated by the enhanced inhibition of bacterial strains and the lower minimum inhibitory concentration values.

## 1. Introduction

The primary objective of pharmaceutical research is to transform an active pharmaceutical ingredient (API) into a solid form with favorable physical and chemical properties [1,2]. However, solid-state drugs face challenges related to poor solubility and low permeability, which hinder their ability to achieve maximum clinical efficacy [3]. Currently, the majority of solid drugs available on the market fall under categories II and IV of the Biopharmaceutics Classification System (BCS), which indicate that their bioavailability is significantly limited due to inadequate water solubility or permeability [4]. Examples include paclitaxel [5], acyclovir [6], and cyclosporine A [7]. Therefore, enhancing the solubility of existing drugs or improving their permeability has become an important aspect of solid-state drug design.

Norfloxacin (NORF, Figure 1), 1-ethyl-6-fluoro-1,4-dihydro-4-oxo-7-(1-piperazinyl)-3-quinoline carboxylic acid [8], is an oral, broad-spectrum fluoroquinolone antibacterial agent [9]. This compound specifically inhibits bacterial DNA. It operates by obstructing DNA replication through its interaction with the gyrase enzyme [10,11]. NORF falls under class IV of the Biopharmaceutical Classification System (BCS) [12,13] and is effective against both Gram-positive and Gram-negative aerobic bacteria. Its primary use is in the treatment of bacterial infections in the lower respiratory and urinary tracts [14,15]. Like many fluoroquinolones, however, NORF faces challenges with water solubility. This issue arises from zwitterionic formation, which is due to proton exchange between the carboxyl group and the piperazine ring, which diminishes its efficacy and bioavailability [16,17]. Currently, efforts to enhance NORF’s solubility have focused on developing various solid preparation forms, such as liposomes [18] and nanoparticles [19]. Nevertheless, high production costs and the limitations of industrialized production have constrained the clinical application of these formulations.

Crystal engineering is an essential tool for the design and synthesis of multicomponent solid forms, such as co-crystals and salts. The NORF molecule has hydrogen-bonding functional groups of COOH and NH. Hence, it can form salts and co-crystals. Pharmaceutical co-crystals are multicomponent crystalline substances formed by an active pharmaceutical ingredient (API) and a co-crystal former (CCF) in the same crystal lattice, according to a fixed stoichiometric ratio [20,21,22]. The selected CCF should have no adverse or toxic pharmacological effects, meet the requirements of generally regarded as safe (GRAS) [23], and have a water solubility superior to that of the API [24]. It is noteworthy that the formation of a salt or a co-crystal depends on the ΔpKa value of the acid–base system. Pharmaceutical salts and co-crystals provide significant opportunities for creating new solid dosage forms that enhance physicochemical properties like solubility, permeability, and bioavailability without altering the pharmacological attributes of the API [25]. For example, the co-crystals prepared with sulfamethazine and p-aminobenzoic acid improved the solubility and antibacterial activity of sulfanilamide [26]; the pharmaceutical co-crystals of ciprofloxacin with carvacrol and thymol showed better antibacterial activity than ciprofloxacin alone [27]; and two co-crystals of olaparib with flavonoids showed significantly increased inhibiting effects on ovarian cancer cells compared to olaparib [28]. Unfortunately, there are only limited reports on the salts/co-crystals of NORF, such as a dual-drug co-crystal with ciprofloxacin [29], a salt co-crystal with saccharin [30], and a co-crystal with riboflavin [31].

The objective of this study was to develop pharmaceutical salts/co-crystals of NORF in order to modulate its solubility and antibacterial activity, motivated by the recent advancements in supramolecular crystal engineering. In this study, we present novel solid forms of NORF with 3,5-dihydroxybenzoic acid (3,5-DBA) and vanillic acid (VA), both of which are pharmaceutical-acceptable phenolic acids (Figure 1). These phenolic acids have been widely utilized in the development of multicomponent solid forms, leading to the improved solubility of various bioactive agents, such as fluconazole [32], 4-aminopyridine [33], penciclovir [34], and gefitinib [35]. Notably, VA is a non-toxic and natural co-crystal former with demonstrated antibacterial activity [36,37], which may enhance the antibacterial effect of NORF.

## 2. Results and Discussion

### 2.1. SCXRD Analysis

The colorless crystal **1** of NORF—3,5-DBA hydrate and crystal **2** of NORF—VA hydrate were successfully synthesized by the solvent evaporation method from methanol/water and acetonitrile/water, respectively. Crystals **1** and **2** are salts [38]. First, according to the ΔpKa rule [39], the ΔpKa of the NH (8.67) of the piperazinyl group of NORF, the COOH (4.04) of 3,5-DBA, and the COOH (4.53) of VA were greater than 3, and the COOH groups of both benzoic acids completely donate their H to the NH on NORF’s piperazine group. This is also verified by the C13-O7 (1.254 Å) and C13-O8 (1.260 Å) in crystal **1**, as well as the C17-O4 (1.256 Å) and C17-O5 (1.255 Å) in crystal **2**. Here, the two C-O bond lengths within the same COO^−^ have comparable values, indicating that the negative charge is almost evenly distributed across the two O atoms of the COO^−^ [40]. The three-dimensional (3D) structure and the intermolecular interactions of the salts were derived from SCXRD data. Detailed crystallographic data and hydrogen bonding configurations for NORF—3,5-DBA hydrate (salt **1**) and NORF—VA hydrate (salt **2**) can be found in Appendix A, respectively.

#### 2.1.1. NORF—3,5-DBA Hydrate (Salt **1**)

NORF—3,5-DBA hydrate crystallizes in the monoclinic system, space group P2_1_/c, with an asymmetric unit consisting of one NORF, one 3,5-DBA, and two water molecules. The stoichiometric ratio of NORF, 3,5-DBA, and water molecules is 1:1:2. There is proton transfer from 3,5-DBA to the piperazine N1 of NORF. A molecule of water acts as a bridge connecting NORF and 3,5-DBA (N1-H1A ··O11, 1.968 Å, 156.46°; O11-H11A···O7, 1.830 Å, 168.23°) (Figure 2a). In addition, an intramolecular hydrogen bond (O6-H6···O5, 1.723 Å, 155.26°) is present with an S(6) ring motif in NORF. Two asymmetric units form the R_4_^4^(12) ring motif by O-H···O hydrogen bonds (O11-H11B···O8, 1.876 Å, 164.96°) (Figure 2b). The ring motifs interact with each other via a N-H···O hydrogen bond (N1-H1B···O12, 1.912 Å, 162.90°), leading to the formation of a one-dimensional (1D) chain (Figure 2c).

The 1D chain, when extended in the reverse direction, assembles into a 2D structure through O-H···O hydrogen bonds (O9-H9···O7, 1.842 Å, 167.28°; O12-H12A···O11, 1.971 Å, 175.95°). This arrangement creates the R_6_^4^(12) ring motif, as illustrated in Appendix A. The 2D structure is connected by a O-H···O hydrogen bond (O10-H10···O6, 2.233 Å, 140.20°) to form a 3D structure (Appendix A). The crystal structure is stabilized with C-H···π (H-to-centroid bond distance: 2.841 Å, 3.106 Å, and 3.531 Å) (Appendix A) interactions between the H on NORF and the benzene ring on 3,5-DBA.

#### 2.1.2. NORF—VA Hydrate (Salt **2**)

NORF—VA hydrate belongs to the monoclinic system, *P*2_1_/c space group. An asymmetric unit is composed of one NORF, one VA, and two water molecules, in a ratio of 1:1:2. The proton on VA transfers to the N atom (N3) of NORF to form a charge-assisted hydrogen bond (N3-H13C···O4, 1.835 Å, 172.20°). Water molecules and VA are connected by O-H···O hydrogen bonds (O8-H8B···O5, 1.725 Å, 168.10°; O9-H9B···O5, 1.871 Å, 170.33°) (Figure 3a). Similar to salt **1**, the NORF molecule features an intramolecular hydrogen bond characterized by an S(6) ring motif (O2-H2···O3, 1.763Å, 154.11°). The asymmetric units form the R_4_^4^(12) ring motif by an O-H···O hydrogen bond (O8-H8A···O4, 2.013 Å, 156.25°) (Figure 3b). The continuous growth in the tetramer forms a 1D chain by O-H···O hydrogen bonding (O9-H9A···O1, 1.938 Å, 158.39°) (Figure 3c).

The 1D chains form a 2D structure by the R_6_^4^(12) ring motif, which is created by N-H···O hydrogen bonds (N3-H13D···O8, 1.875 Å, 169.01°) (Appendix A). The reverse 2D structures form a 3D structure through O-H···O hydrogen bonds (O7-H7···O9, 1.887 Å, 146.40°) (Appendix A). There are two π-π interactions (3.579 Å and 3.695 Å) between NORF molecules. Additionally, the H15A on the piperazine ring interacts with the benzene ring of VA (2.589 Å) (Appendix A).

### 2.2. PXRD Analysis

The purity and homogeneity of active pharmaceutical ingredients (APIs) and their crystalline solid forms are predominantly evaluated by PXRD. This method involves comparing the unique diffraction patterns from the initial materials with those from binary or multicomponent products. PXRD is both quick and reliable for verifying the existence of crystalline phases [41]. The PXRD data of the three starting components, NORF, 3,5-DBA, and VA, as well as the two salts, were gathered and are given in Figure 4. As shown in Figure 4a, the characteristic diffraction peaks of NORF at 2θ of 7.87° and 30.35°, and those of 3,5-DBA at 2θ of 6.82° and 27.37°, are absent in the pattern of salt **1**. Instead, new peaks for salt **1** appear at 2θ of 13.94° and 26.32°, indicating the formation of a new solid form. Similarly, as depicted in Figure 4b, the characteristic diffraction peaks of NORF at 2θ of 7.19°, 12.10°, and 32.50°, along with VA at 2θ of 9.20°, 12.77°, and 21.89°, are not present in the pattern of salt **2**. New peaks for salt **2** emerge at 2θ of 6.49°, 13.24°, and 13.71°. Additionally, the patterns of the powder samples closely align with the simulated patterns derived from single-crystal diffraction data, further confirming the formation of a new phase with high purity.

### 2.3. Thermal Analysis

The thermal behavior of NORF, CCFs, and salts was analyzed by DSC and TGA. The DSC curves for the NORF experiments with new solid phases are shown in Figure 5. It shows that the two salts exhibited a distinct endothermic peak with a pronounced profile, and their peak temperatures were different from those of NORF or the coformers. This change meant that new phases were formed. For salt **1**, its peak temperature (281.1 °C) was higher than those of NORF (221.6 °C) and 3,5-DBA (237.8 °C). However, the peak temperature (218.0 °C) of salt **2** was between those of NORF (221.6 °C) and VA (211.6 °C). This shows that the melting point of the salt has little relationship with the API and CCF. Furthermore, endotherm peaks prior to melting were observed for the two salts, indicating that the two salts were of hydrous nature, which was also supported by their single-crystal structures. The release of water over a wide temperature range suggests a variety of hydrogen-bonding interactions and energies involved in the bonding of water to the salts [42]. The stoichiometry of the hydrate forms was verified through TGA (Figure 6). For salt **1** and salt **2,** the weight loss in the first stage started from approximately 103 °C and 110 °C, and the total weight losses were 7.08% and 6.89%, respectively. The weight losses were all consistent with the stoichiometric ratio of water molecules.

### 2.4. FT-IR Analysis

FT-IR spectra are useful for identifying changes in hydrogen bonding during co-crystallization by analyzing the frequency shifts in the relevant vibrational bands [43]. IR spectroscopy was used to analyze the two novel salts, along with pure NORF, 3,5-DBA, and VA, and the obtained spectra are presented in Appendix A. The characteristic peaks of NORF at 1622 and 1729 cm^−1^ were attributed to the C=O stretching of pyridinone and the C=O stretching of carboxylic acids, respectively, aligning with the findings reported in the existing literature [42,44]. Both salt **1** and salt **2** exhibited characteristic peaks for pyridinone C=O at 1625 cm^−1^ that were essentially consistent with NORF. Furthermore, in the spectra of salt **1** and salt **2**, the absorption peaks for the C=O stretching vibration of carboxylic acids shifted from 1729 to 1700 cm^−1^ and 1713 cm^−1^, respectively. These significant wavenumber changes may have resulted from intramolecular interactions (hydrogen bonding), consistent with the results obtained by SCXRD. This suggested that the ketone and carboxylic acid group of NORF remained stable without forming zwitterions. Additionally, the broad IR absorption frequencies between 2480 and 2520 cm^−1^ for the two salts indicate the protonation of the N atom (NH_2_^+^) in the piperazinyl ring, with the proton clearly transferring from the carboxylic acid. This observation further implies an interaction between NORF and phenolic acid via the piperazine ring [45]. These changes were corroborated by the crystal structure analysis.

### 2.5. Equilibrium Solubility Study

Solubility is a critical parameter for enhancing the bioavailability and effectiveness of a drug product [46]. The forming salt is utilized to increase the solubility of insoluble drugs and is considered one of the effective methods for increasing their aqueous solubility [47]. To assess whether the salt technique improved the solubility of NORF, we conducted tests of the solubility of NORF and the two salts in dilute hydrochloric acid solution (pH = 1.2), phosphate buffer (pH = 6.8), and pure water, separately. A slight excess of the appropriate NORF and the two salts’ solid form was dissolved in each medium and stirred at 100 rpm at 37 ± 0.5 °C for 24 h. The solution was filtered, and the residue from the filtration was used to measure the NORF concentration by HPLC, with the results shown in Figure 7. As depicted in Figure 7, it was observed that the solubilities of both salts decreased with increasing pH, which was consistent with the trend in the solubility changes for NORF. Although both the pure NORF and the two salts displayed similar solubility, it was consistently found that NORF in the salt exhibited a higher solubility than pure NORF when placed into the corresponding buffer solutions, following the orders: salt **2** > salt **1** > NORF. Consequently, it can be concluded that the concentrations of NORF from these two solid forms in media with pH 1.2, pH 6.8, and pure water are 1.6–3.9 times higher than those of pure NORF, indicating an advantage provided by utilizing a salt strategy to establish superior solubility. Noteworthily, the pH 6.8 environment chosen for this study accurately reflects the physiological conditions of the human small intestine. Consequently, the enhanced solubility contributes to facilitating better absorption, which lays a foundation for further improving the bioavailability of NORF.

### 2.6. In Vitro Powder Dissolution Study

This study systematically discussed the dissolution behavior of NORF and the two salts in salivary pH 1.2 (dilute hydrochloric acid solution), pH 6.8 (phosphate buffer), and pure water. The results are displayed in Figure 8, providing a reference value for evaluating the dissolution behavior of orally ingested drugs. Within 360 min, the dissolution of the two salts was slightly improved compared to that of the pure drug in simulated gastric fluid. This can be attributed to NORF being an amphoteric molecule with both acidic (COOH) and basic (NH) functional groups, resulting in good solubility under acidic conditions. However, in pure water and a buffer solution of pH 6.8, both salts showed better dissolution properties than NORF. Particularly in the pure water medium, the dissolution (>80%) of both salts in this study was significantly higher than that of NORF (~36%) at 360 min. Compared to that of NORF, the dissolution of the two salts was significantly improved in pure water. Powder dissolution analysis indicated that salt **1** and salt **2** presented a better physiochemical property than NORF, potentially leading to improved bioavailability.

### 2.7. Antibacterial Activity Study

The improvement in pharmacological activities such as antibacterial and anticancer effects through co-crystallization [26,28,48] has been reported in the literature. To obtain quantitative and precise data on the biological activity of the current salts, minimum inhibitory concentration (MIC) tests were conducted against bacterial strains such as *Escherichia coli* (*E. coli*), *Pseudomonas aeruginosa* (*P. aeruginosa*), *Bacillus subtilis* (*B. subtilis*), and *Staphylococcus aureus* (*S. aureus*). The results are summarized in Table 1, which clearly indicates that the MIC values for both salts are significantly lower than those for NORF, showing enhanced antimicrobial susceptibility. The salts demonstrated moderate to relatively high effectiveness against all tested pathogens, with MIC values ranging from 0.12 to 0.5 µg/mL. Prior research has also suggested that dissolution performance is a crucial factor in determining antibacterial activity, with samples that dissolve more rapidly exhibiting stronger antibacterial effects [49]. Therefore, both salts may have enhanced antibacterial activity owing to their increased solubility.

## 3. Materials and Methods

### 3.1. Materials

NORF (C_16_H_18_FN_3_O_3_, purity > 99.0%) was purchased from Anhui Ze Sheng Science and Technology Co., Ltd. (Shanghai, China), while 3,5-DBA (C_7_H_6_O_4_, purity > 99%) and VA (C_8_H_8_O_4_, purity > 98%), as well as acetonitrile (HPLC-grade), methanol (HPLC-grade), HCl (37%), and KH_2_PO_4_ were purchased from Anhui Ze Sheng Science and Technology Co., Ltd. (Shanghai, China) or Shanghai Bide Pharmaceuticals Science and Technology Co., Ltd. (Shanghai, China). All chemicals were used without a purity test or further processing, and distilled water was purchased from a local market.

### 3.2. Synthesis of Salts

#### 3.2.1. NORF—3,5-DBA Hydrate (1:1:2) Salt **1**

NORF (31.9 mg, 0.1 mmol) and 3,5-DBA (30.8 mg, 0.2 mmol) were dissolved in 13 mL of acetonitrile/methanol (12:1, *v*/*v*). This solution was thoroughly stirred in a water bath maintained at 60 ± 1 °C for 3 h. After cooling to room temperature, the solution was filtered, and the filtrate was allowed to evaporate slowly. One week later, colorless crystals were obtained, with a yield of 64% based on NORF.

#### 3.2.2. NORF—VA Hydrate (1:1:2) Salt **2**

NORF (31.9 mg, 0.1 mmol) and VA (33.6 mg, 0.2 mmol) were dissolved in 27.5 mL of acetonitrile/distilled water (10:1, *v*/*v*). This solution was thoroughly stirred in a water bath maintained at 60 ± 1 °C for 3 h. After cooling to room temperature, the solution was filtered, and the filtrate was allowed to evaporate slowly. One week later, colorless crystals were obtained, with a yield of 65% based on NORF.

### 3.3. Single-Crystal X-ray Diffraction (SCXRD)

The single crystals were tested on a Bruker D8 Quest diffractometer (Bruker, Billerica, MA, American) using Mo Kα radiation (λ = 1.54178 Å) at room temperature. The structures were solved by direct methods using OLEX2 software (version 1.5) [50] and were refined on *F*^2^ by the full-matrix least-squares technique using the SHELXL program (version 1.0.1678) [51]. Refinement of non-hydrogen atoms was performed through anisotropic displacement parameters. Hydrogen atoms were placed in calculated positions and refined with a riding model. Crystal structures were viewed and analyzed using DIAMOND 4 software (version 3.2) [52]. The crystallography data and hydrogen-bonding interactions were analyzed and are summarized in Appendix A, respectively. The crystallographic data of salt **1** and salt **2** were submitted to the Cambridge Crystallographic Data Centre (CCDC no. 2377031-2377032).

### 3.4. Powder X-ray Diffraction (PXRD)

The PXRD powder patterns were analyzed using a Bruker D8 Advance diffractometer (Bruker, Bremen, Germany) equipped with Cu Kα radiation (λ = 1.5406 Å) operating at 40 kV and 15 mA. XRD patterns were recorded across a 2θ range of 5–50° with a scan rate of 5°/min. All measurements were conducted at ambient temperature. Mercury software (version 2023.2.0) [53] was used to simulate diffraction patterns from single-crystal X-ray diffraction data.

### 3.5. Differential Scanning Calorimetry (DSC)

The DSC experiments were conducted using a DSC 204 F1 instrument (Netzsch, Selb, Germany) under a nitrogen atmosphere with a gas flow of 20 mL/min and a constant heating rate of 10 °C/min. Each test utilized approximately 5–10 mg of the sample.

### 3.6. Thermogravimetric Analysis (TGA)

Thermogravimetric analysis was carried out using a TGA STA2500 instrument (Netzsch, Selb, Germany). Nitrogen was used as the purge gas at a flow rate of 20 mL/min. The sample was placed in an open aluminum oxide pan and heated at a rate of 10 °C/min from 30 °C to 600 °C.

### 3.7. Fourier-Transform Infrared Spectrophotometry (FT-IR)

FT-IR studies were conducted using an IRAffinity-1 instrument (Shimadzu, Nakagyo-ku, Japan) across a spectrum range of 4000–400 cm^−1^, with a resolution of 4 cm^−1^ and 32 scans. The KBr pellet method was employed, where samples were weighed and mixed with KBr in a specific ratio (1:100, *w*/*w*) before being compressed into tablets for analysis. The scanning frequency was set at 32 times, and the resolution was maintained at 4 cm^−1^.

### 3.8. High-Performance Liquid Chromatography (HPLC)

The concentration of NORF in solution was measured using a C18-WR column (250 × 4.6 mm, 5 μm particle size) on an LC-20 AD liquid chromatography system (Shimadzu, Japan) at 278 nm. The mobile phase was composed of a 0.025 mol/L phosphoric acid aqueous solution (pH adjusted to 3.0 ± 0.1 with triethylamine) and acetonitrile in a ratio of 87:13 (*v*/*v*). The flow rate was maintained at 1 mL/min and the temperature at 30 °C. The retention time for NORF was approximately 26 min, which was used to determine its concentration in the solution.

### 3.9. Equilibrium Solubility Measurement

The equilibrium solubility of NORF and the two salts was determined in three dissolution media using the excess powder dissolution method [54]. An excess amount of NORF and a NORF-equivalent amount of salts were added to 100 mL of hydrochloric acid solution (pH 1.2), phosphate buffer solution (pH 6.8), and distilled water. Samples were stirred at 100 rpm at a temperature of 37 ± 0.5 °C for 24 h, then filtered, appropriately diluted, and analyzed using HPLC. The saturation solubility determination experiments were conducted in triplicate to ensure consistency and reliability of the results.

### 3.10. In Vitro Powder Dissolution

According to the Chinese Pharmacopoeia paddle method, the powder was dissolved on an RC806ADK dissolution instrument (Tianjin, China) in 250 mL of buffer at pH 1.2 (simulated gastric fluid), pH 6.8 (simulated small intestine) and distilled water at 37 ± 0.5 °C. In the experiment, the stirring speed was 100 rpm/min. NORF, salt **1**, and salt **2** were ground and passed through a 200-mesh sieve. For the powder dissolution experiments, 40 mg of NORF or an equivalent amount of NORF salt was weighed. Samples of 1 mL were extracted at specified time intervals of 15, 30, 45, 60, 90, 120, 180, 240, 300, and 360 min. After each extraction, an equal volume of dissolution medium was added back to maintain a constant volume within the dissolution system. The samples were then filtered, and the percentage of drug release was analyzed using HPLC. All experiments were conducted in triplicate.

### 3.11. Antibacterial Activity Measurement

*Pseudomonas aeruginosa* (*P. Aeruginosa*), *Staphylococcus aureus* (*S. aureus*), *Escherichia coli* (*E. coli*), and *Bacillus subtilis* (*B. subtilis*) were chosen as experimental strains to comparatively investigate the antibacterial effects of the salt and NORF. The minimal inhibitory concentration (MIC) was determined using the dilution method. In aseptic conditions, all test samples were diluted to gradient concentrations with medium and inoculated with approximately 1 × 10^8^ cfu/mL of actively dividing bacterial cells. After incubation at 37 °C for 24 h, bacterial growth was assessed visually and spectrophotometrically. The MIC was defined as the lowest concentration that inhibited bacterial growth. All the experiments were conducted in triplicate.

## 4. Summary

In this study, salts of the antibacterial agent NORF with phenolic acids, including 3,5-DBA and VA, were meticulously designed and successfully self-assembled based on established optimization tactics and synergistic enhancement efficacy. The obtained salts were structurally characterized, and their in vitro biopharmaceutical peculiarities were systematically investigated both theoretically and experimentally. The solubility experiments indicated that the salts could enhance the solubility of NORF and improve the release behavior. Furthermore, the antimicrobial activity tests proved that salts exhibit a satisfactory increase in antimicrobial effect, pointing to a novel direction for optimizing the efficacy of NORF. Clearly, in vivo confirmations are required to exploit their applications in the field of pharmacy.

## Figures and Tables

**Figure 1 antibiotics-13-00888-f001:**
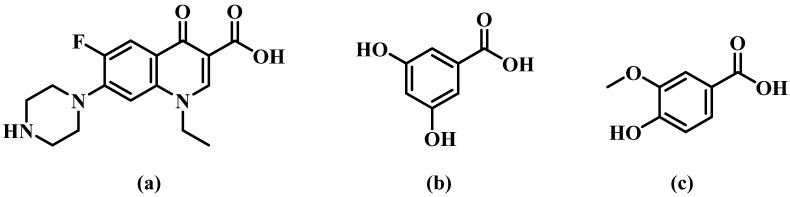
Chemical structures of (**a**) norfloxacin (NORF, MW = 319.33 g/mol), (**b**) 3,5-dihydroxybenzoic acid (3,5-DBA, MW = 154.12 g/mol), and (**c**) vanillic acid (VA, MW = 168.14 g/mol).

**Figure 2 antibiotics-13-00888-f002:**
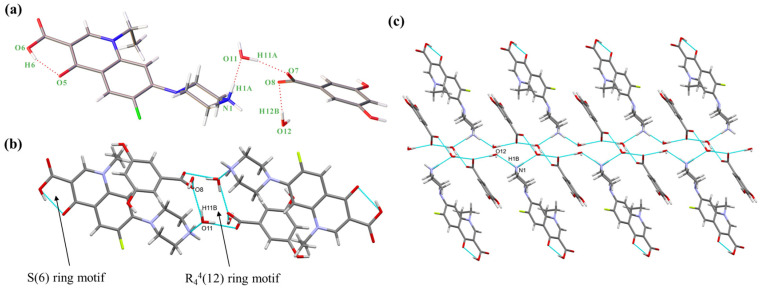
(**a**) Asymmetric unit of salt **1**; (**b**) cyclic hydrogen bond ring motif consisting of two 3,5-DBA and two water molecules; (**c**) 1D chain of salt **1**. Both the red solid lines and the light blue dashed lines represent hydrogen bonds.

**Figure 3 antibiotics-13-00888-f003:**
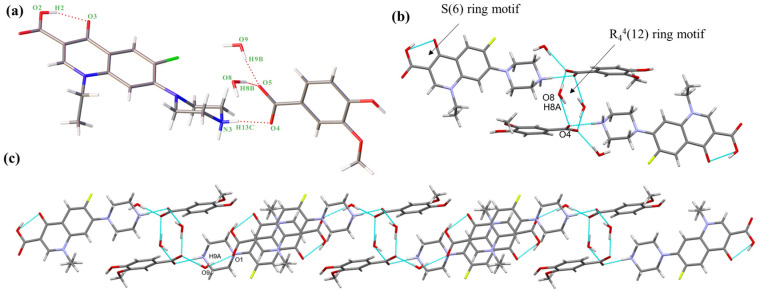
(**a**) Asymmetric unit of salt **2**; (**b**) cyclic hydrogen bond ring motif consisting of two VA and two water molecules.; (**c**) 1D chain of salt **2**. Both the red solid lines and the light blue dashed lines represent hydrogen bonds.

**Figure 4 antibiotics-13-00888-f004:**
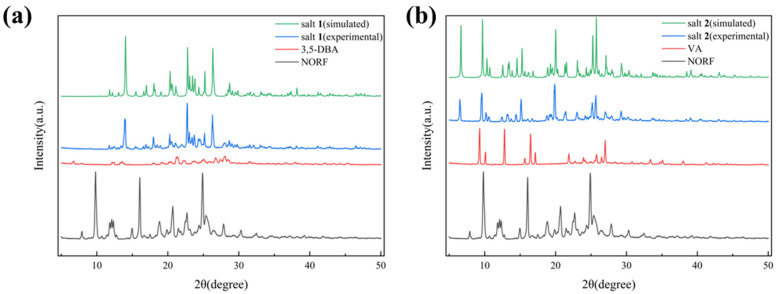
Comparison of experimental and simulated PXRD patterns of (**a**) NORF, 3,5-DBA, and salt **1**; (**b**) NORF, VA, and salt **2**.

**Figure 5 antibiotics-13-00888-f005:**
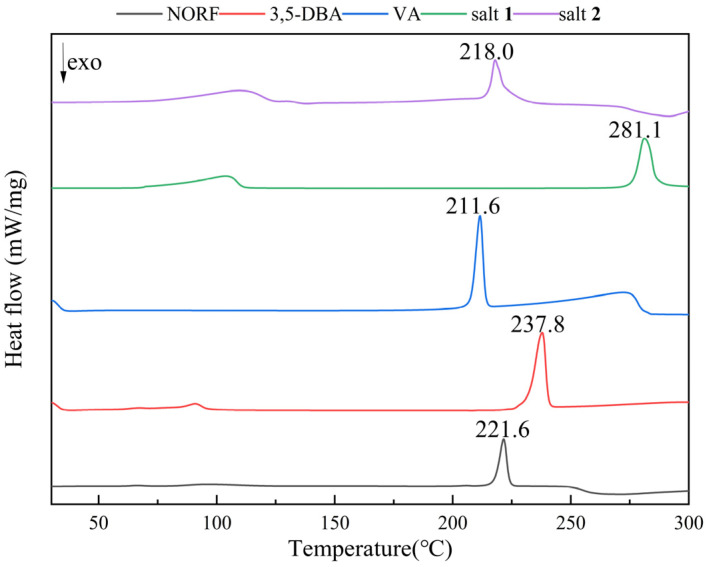
DSC curves of NORF, 3,5-DBA, VA, salt **1,** and salt **2**.

**Figure 6 antibiotics-13-00888-f006:**
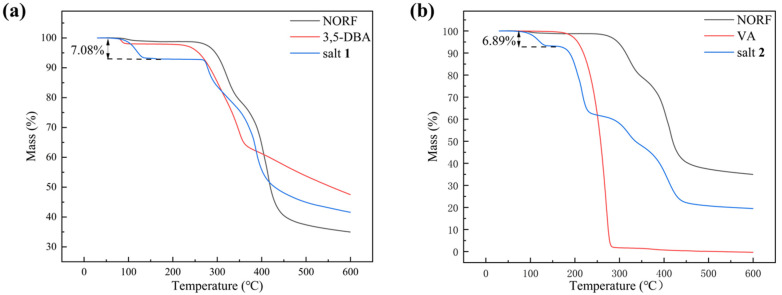
TGA curve of (**a**) NORF, 3,5-DBA, and salt **1**; (**b**) NORF, VA, and salt **2**.

**Figure 7 antibiotics-13-00888-f007:**
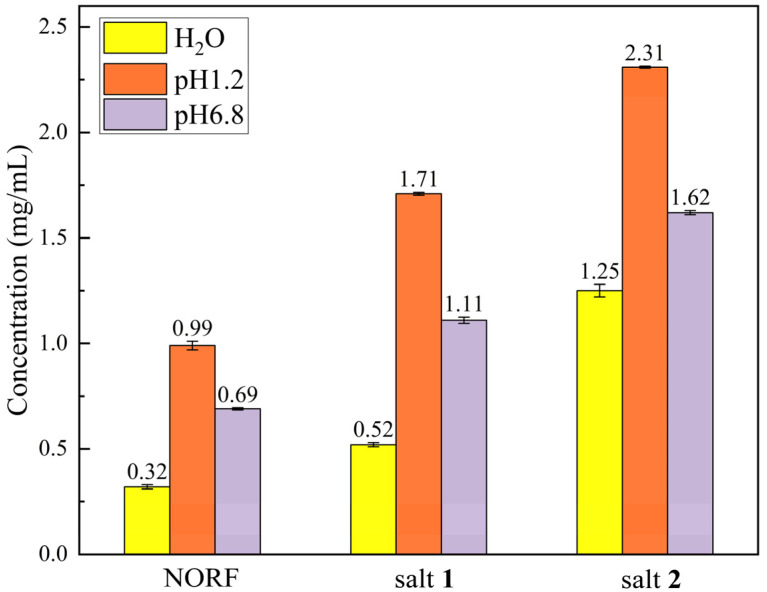
Equilibrium solubility of NORF and its salts.

**Figure 8 antibiotics-13-00888-f008:**
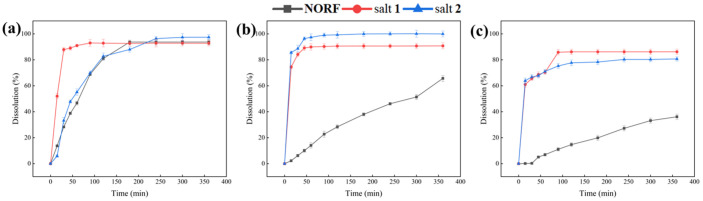
Dissolution profiles of NORF, salt **1**, and salt **2** in buffer (**a**) pH 1.2, (**b**) pH 6.8, and (**c**) pure water.

**Table 1 antibiotics-13-00888-t001:** Minimum inhibitory concentration values for NOX, CCFs, and salts.

MIC (μg/mL)	*E. coli*	*P.* *aeruginosa*	*B.* *subtilis*	*S. aureus*
NORF	0.5	0.5	0.5	0.5
3,5-DBA	512	512	512	512
VA	1024	1024	1024	1024
salt **1**	0.5	0.25	0.25	0.125
salt **2**	0.25	0.25	2	0.12

## Data Availability

Data are contained within this article and the Appendix A.

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
