# Peer review of "Two Novel Hydrate Salts of Norfloxacin with Phenolic Acids and Their Physicochemical Properties"

_antibiotics, 2024, doi:10.3390/antibiotics13090888_

Round 1
Reviewer 1 Report
Comments and Suggestions for Authors
1. The quality of structure 2 is marginal based on the value of Rint. I recommend finding a better crystal and redoing the structure.
2. In the caption for Fig. 8 it should be "NORF" not "NOX".
3. In line 238, "parameter" (singular).
4. Lines 234-237: An incomplete sentence. Please correct.
5. Lines 343-346: Which method was used, "shaking" or "stirring". Please correct.
6. The final section is a "summary" not "conclusions". If it were to be the latter, it has to contain a thoughtful discussion of the results, not just a listing of them.
7. I don't see that the Hirshfeld surface analysis is particularly relevant to the rest of the study unless it can be shown to bear strongly on the question of increased solubility for the co-crystals. Otherwise, it is just "padding".

Generally good. A few missing indefinite articles (the, a, an, etc.).
Reviewer 2 Report
Comments and Suggestions for Authors
I think that since this journal is a biochemical journal and not a crystal chemical one, a detailed analysis of the Hirschfeld surfaces is not necessary. In any case, this description, including Fig. 4, can be moved to the Supporting Information without detriment to the manuscript.
Reviewer 3 Report
Comments and Suggestions for Authors
The manuscript describes the preparation of two cocrystals containing norfloxacin (NORF) in the presence of two different coformers with 3,5-dihydroxybenzoic acid (3,5-DBA) and vanillic acid (VA). From this, I want to make some observations:
1. The title should be changed since points 2.1.1 and 2.1.2 mention a proton transfer (of the acid-base type), indicating that they are salts, not cocrystals [1].
2. Observing the differences between two C−O bond distances, particularly COOH/COO- with single crystal X-ray diffraction data, is recommended to discern whether the two compounds are salts [2].
3. Applying deltapKa's rule of thumb is recommended to differentiate between a salt, cocrystal, or salt-cocrystal continuum. This approach will provide a clear understanding of the compound type [3].
4. Suppose authors use the Graph Set Patterns of Hydrogen Bondings. In that case, it is recommended that you do so with all because figures 2b and 3b show intramolecular hydrogen-bonded patterns (S) that these can describe [4].
5. Regarding the pi-pi stackings described in Figures S1-2, one must consult the geometric parameters to consider whether such an interaction is genuine. Cg...Cg values ​​of 3.9 Šor greater can no longer be considered pi-pi stacking. All the parameters described in the article should be used to define whether it is a stack [5].
It is recommended that authors make these major corrections so that the manuscript is suitable for publication.
1. Da Silva, C.C.; Guimarães, F.F.; Ribeiro, L.; Martins, F.T. Salt or Cocrystal of Salt? Probing the Nature of Multicomponent Crystal Forms with Infrared Spectroscopy. Spectrochim. Acta - Part A Mol. Biomol. Spectrosc. 2016, 167, 89–95, doi:10.1016/j.saa.2016.05.042.
2. Garai, A.; Biradha, K. Cocrystals and Salts of 4,4′-Dinitro-2,2′,6,6′-Tetracarboxybiphenyl with N-Heterocycles: Solid State Photodimerization of Criss-Cross Aligned Olefins and Photophysical Properties. Cryst. Growth Des. 2020, 20, 8059–8070, doi:10.1021/acs.cgd.0c01305.
3. Childs, S.L.; Stahly, G.P.; Park, A. The Salt−Cocrystal Continuum: The Influence of Crystal Structure on Ionization Stat. Mol. Pharm. 2007, 4, 323–338, doi:10.1021/mp0601345.
4. Bernstein, J.; Davis, R.E.; Shimoni, L.; Chang, N.-L. Patterns in Hydrogen Bonding: Functionality and Graph Set Analysis in Crystals. Angew. Chem. Int. Ed. 1995, 34, 1555–1573, doi:10.1002/anie.199515551.
5. Jaime-Adán, E.; Germán-Acacio, J.M.; Páez-Franco, J.C.; Lara, V.H.; Reyes-Marquez, V.; Morales-Morales, D. Exploring the Persistence of the Fluorinated Thiolate 2,3,5,6-S(C6F4H-4) Motif to Establish ΠF-ΠF Stacking in Metal Complexes: A Crystal Engineering Perspective. Dalt. Trans. 2024, doi:10.1039/d4dt01978d.
Round 2
Reviewer 3 Report
Comments and Suggestions for Authors
The changes made to the manuscript address the observations made in the first revision.
Therefore, the manuscript is ready to be published in the journal.
Author Response
Thanks for your suggestions.